# From fangs to antidotes: A scoping review on snakebite burden, species, and antivenoms in the Eastern Mediterranean Region

**Ali Alshalah**[1]*, **David J. Williams**[2], **Alessandra Ferrario**[1]*

**1** Information Systems for Health Unit, Department of Science, Information and Dissemination, WHO Regional Office for Eastern Mediterranean, Cairo, Egypt, **2** Regulation and Prequalification Department, World Health Organization, Geneva, Switzerland

* Ali-Haydar-Hashim.Al-Shalah1@alumni.lshtm.ac.uk (AA); aferrario@who.int (AF)

## Abstract

### Background

Snake bites cause considerable morbidity and mortality worldwide, yet evidence from low- and middle-income countries remains fragmented. This is particularly the case in Eastern Mediterranean Region where available data on snake bites is relatively weak. Without reliable data, it is difficult to make the case for greater visibility and investment to address the snakebite burden in this Region. A scoping review was therefore conducted to summarize evidence on snake bites in countries of the Eastern Mediterranean.

### Methodology/principal findings

The review employed manual and electronic searching methods of four databases plus Google Scholar, ultimately including 196 records from 20 countries published between 2000 and 2023. More than half originated from Iran, Morocco, and Pakistan. Many records lacked information on permanent sequalae, disability, snake species, and types and sources of antivenoms. When identified, offending snakes belonged to 30 species. Use of more than 12 types of antivenoms were described across the Region, and some were not specific to indigenous species.

### Conclusion/significance

Despite the relatively large number of publications identified, the data were concentrated in just a few countries in the Region, and there was little or no information available for the remainder. As is the case worldwide, disability associated with snake bites was poorly characterized and quantified across the Region. There is an urgent need for concrete action at national and regional levels to enhance epidemiological surveillance, research, and the collection of clinical, disability and outcomes data to inform policy and public health investment. Greater regional cooperation and collaboration is also crucial for addressing this neglected disease throughout the Region.

WHO logo is not permitted. This notice should be preserved along with the article's original URL.

**Data Availability Statement:** All relevant data are within the manuscript and its Supporting Information files.

**Funding:** The author(s) received no specific funding for this work.

**Competing interests:** The authors have declared that no competing interests exist.

## Author summary

Snake bites are a particular public health concern worldwide. Compared to other parts of the world, the available data on snake bites in the Eastern Mediterranean Region–a region spanning large parts of North Africa and the Middle East all the way to Pakistan and Afghanistan–is relatively weak. Without reliable data it is difficult to make the case for greater visibility and investments to address snakebite burden in the Region. To address this knowledge gap, we conducted the first scoping review to summarize evidence on snake bites from all 22 countries of the Eastern Mediterranean. Despite identifying 196 publications, we found that evidence was concentrated in a few countries and remained limited or absent for most of the Region. Moreover, a significant proportion of publications lacked information on long-term health consequences, snake species involved, and types and sources of antivenoms used. Our review highlights the need to improve data collection on snakebite burden as part of routine health information systems and community-based surveillance, and to address disability in future research. Additionally, findings suggest that some countries need to review their procurement practices of antivenoms to ensure availability and use of appropriate quality products in the future.

## 1. Introduction

It is estimated that snake bites cause around 81,000–138,000 deaths and 400,000 permanent disabilities globally every year [1] with research putting years of life lost due to snakebite envenoming at 38 per 100,000 and 2.9 million in absolute numbers annually worldwide [2]. However, such estimates are based on extrapolation from national health systems data or small surveys at subnational levels, and may fail to capture the true burden of the condition [3].

Snakebite envenoming exhibits sociodemographic and geographic disparity in burden. It disproportionally affects vulnerable populations like rural communities, working children, people living in poverty [4], and crisis-affected populations [5]. The burden also varies significantly worldwide [6] with South and Southeast Asia and sub-Saharan Africa experiencing considerable morbidity and mortality [7]. This regional variability has been attributed to differences in human and snake ecology, socioeconomic vulnerability, health system factors, health-seeking behavior, and humanitarian crises [2,5].

Antivenoms are considered the mainstay of management of snakebite envenoming [8,9]. They are most often produced from the purified blood of venom-immunized horses and sheep [10,11]. Prompt administration of antivenoms after envenoming is paramount [12] with research indicating that in some settings each one-hour delay would increase mortality by approximately 1% [6,13]. Notwithstanding their importance, the global supply of these treatments has been plagued by shortages and disruptions [14]. Antivenom production can be logistically complex and relatively costly [10,11], and some products require efficient cold-chain systems for storage and transportation which are difficult to attain in affected low-resource areas [15,16]. Further exacerbated by major manufacturers ceasing production, the global supply of antivenoms presently lies below one-third of estimated demand and potentially contributes to more than 100,000 deaths annually [17].

In 2017, the World Health Organization (WHO) added snakebite envenoming to its list of neglected tropical diseases, a step that was described as a "milestone" in terms of enhancing visibility and funding to address this disease [18]. This was followed by the 2018 resolution (WHA71.5) on snakebite envenoming [19] and the 2019 release of WHO strategy setting a

target to reduce the number of deaths and cases of disability by 50% by 2030 [4]. The strategy comprised four strategic objectives: empower and engage communities; ensure safe, effective treatment; strengthen health systems; and increase partnerships, coordination, and resources [4]. Although WHO's strategy has entered its fifth year, research predicts that the goal to halve mortality by 2030 might not be met [2].

Compared to other parts of the world, the available data for the Eastern Mediterranean Region (EMR) is relatively weak with regards to snake bites [20]. Stretching across large parts of North Africa and the Middle East all the way to Pakistan and Afghanistan, this region is home to nearly 767 million people [21], and encompasses 22 countries and territories [22] (shown in Fig 1). Without reliable data, it is difficult to make the case for greater visibility and investment to address snakebite burdens in EMR countries. To address this, we undertook a scoping review of available data to inform the development of recommendations on how to integrate data on snake bites into routine health information systems in EMR countries. Our primary research question was: "Which data are collected and published on snakebite burden, species and antivenoms in countries of the WHO Eastern Mediterranean Region?"

## 2. Methods

A scoping review design was chosen to summarize and map available evidence and to highlight existing gaps in the literature [23]. The review was guided by the work of Arksey and O'Malley [24], Levac et el. [23], and the Preferred Reporting Items for Systematic Reviews and Meta-Analyses extension for Scoping Reviews (PRISMA-ScR) checklist [25] provided in S1 Supplement.

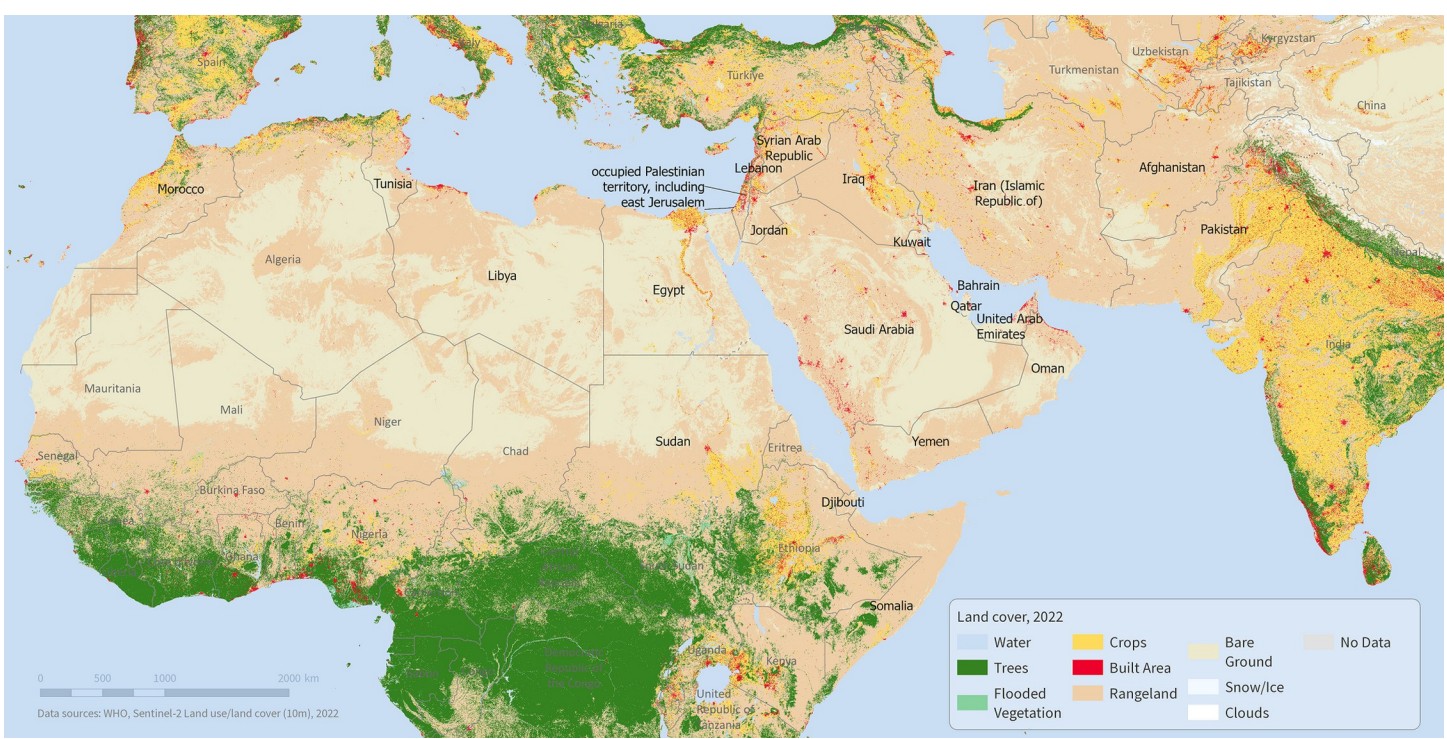

**Fig 1. MAP OF THE EASTERN MEDITERRANEAN COUNTRIES.** Note: The designations employed and the presentation of the material in this publication do not imply the expression of any opinion whatsoever on the part of WHO concerning the legal status of any country, territory, city or area or of its authorities, or concerning the delimitation of its frontiers or boundaries. Dotted and dashed lines on maps represent approximate border lines for which there may not yet be full agreement. Data source for the map: WHO and Sentinel-2 10m land use/land cover time series of the world. Produced by Impact Observatory, Microsoft, and Esri https://www.arcgis.com/home/item.html?id=cfcb7609de5f478eb7666240902d4d3d.

The review protocol was not registered but it is available from authors. The review was carried out over the course of six months from January to July 2023.

## 2.1. Search strategy

Our search combined electronic and manual searching methods to capture published journal articles, conference papers, and grey literature. We consulted four electronic databases: PubMed, Web of Science, Index Medicus for the Eastern Mediterranean (IMEMR), and CINAHL. We treated Google Scholar as part of our manual searching strategy owing to its limitations, namely its 256-characters search limit and maximum display of first 1000 records without specification of ordering [26]. We formulated a search strategy in consultation with a specialized librarian. The strategy had two central concepts: snake bites and EMR countries and incorporated a combination of MeSH terms and text words. We tested, refined, and tailored the strategy to match each included database (provided in S2 Supplement). The search was executed on three databases from February 13 to 14, 2023, and expanded on April 6, 2023, to include the Index Medicus for the Eastern Mediterranean Region (IMEMR).

Manual searching methods included snowballing of search-captured literature reviews; expert consultation to locate further articles and data sources; searching websites of national Ministries of Health to obtain grey literature e.g., the most recent statistical health report containing data on snake bites; and utilizing Google Advanced Search tool with English, Arabic, and French keywords.

Our search was limited to publications from 2000 to 2023 on the premise that older articles may be outdated relative to present data sources.

## 2.2. Study selection

We exported records to the citation management software (Zotero 6.0) so that duplicates could be removed electronically and checked manually.

We included only primary records involving confirmed snakebite human victims regardless of envenoming status. We therefore excluded cross-sectional studies on knowledge, perspectives, and experiences; *in vivo* and *in vitro* studies; modeling studies; and literature reviews. Furthermore, we limited inclusion to articles published in Arabic, English, French, and Persian since they are the major publishing languages of health research in EMR [27]. Inclusion and exclusion criteria are further specified in Box 1 below.

One reviewer first performed screening of titles and abstracts against our inclusion and exclusion criteria. If the abstract was not available, the article was included in the subsequent stage. Relevant records then underwent full-text retrieval and screening. Weekly meetings with a second researcher took place to discuss and resolve emerging uncertainties during this stage.

We contacted 22 authors and journals via email to (i) request articles that could not be retrieved, or to (ii) make enquiries if information was missing e.g., the country of cases was not specified. All non-respondents were sent second follow-up emails, and the final response rate was 50% (11/22).

## 2.3. Data charting, synthesis, and reporting

Data charting took place via a standardized Excel extraction form. The process was largely iterative i.e., some data items were later added [28], and multi-staged. One reviewer first extracted study characteristics from all but Persian studies as another reviewer was sought for those. The reviewers collected data on country focus, language, date of publication, journal/conference name, type of study as per Bolon et al. [29] with adaptations, study aim, first author, and

BOX 1. INCLUSION AND EXCLUSION CRITERIA.

**Inclusion criteria**:

1. Published and grey literature addressing snake bites in humans.

2. Concerning at least one of the 22 countries of the WHO EMR.

3. Reported in English, French, Arabic or Persian.

4. Published between January 2000 and February 2023.

**Exclusion criteria**:

1. Publications not involving snakebite victims, including biochemical, pharmacological, and toxicological studies (i.e., in vivo and in vitro); modeling studies; and literature reviews.

2. Publications addressing snake bites in animals or bites by other animals (e.g., scorpions) including when data cannot be extracted for snake bites.

3. Cross-sectional studies on knowledge, perceptions, and attitudes.

4. Concerning snake bites outside EMR countries, including global/regional studies if the data cannot be extracted for EMR countries.

5. Reported in languages other than English, French, Arabic and Persian.

6. Published outside the timeframe between January 2000 and February 2023.

academic affiliations of authors. Second, reviewers collected and organized data numerically and thematically around three a priori pillars. The first pillar was the burden of snake bites and encompassed total number of cases reported, number of cases included in the analysis, deaths, permanent sequelae, disability, sociodemographic profile of cases, and national-level incidence and case fatality. Second pillar was snake species i.e., offending species and snake identification methods. For this, one reviewer cross-referenced offending species from older publications against recent taxonomic revisions to ensure consistency in species identification. Further, we compared the species reported in the literature with WHO's Snakebite Information and Data Platform [https://snbdatainfo.who.int/] and herpetology and toxicology guides on the Region [30–32]. The third pillar covered types and manufacturers of antivenoms used, specificity (corroborated with WHO's platform), number of cases receiving antivenom, and related antivenom challenges. A second reviewer was sought in case of uncertainty.

The quality of included studies was not appraised, in following Arksey and O'Malley framework [23]. We used frequencies, percentages, tables, and descriptive narrative summaries to report findings.

## 3. Results

The search yielded 1489 records in total. Following deduplication, 1039 original records were retained and screened by title and abstract. Of these, 743 were deemed irrelevant whilst 296 were eligible for full-text retrieval and screening. Subsequently, 90 articles were excluded, 10 could not be retrieved, and 196 were ultimately included in the review. The search and its

results are displayed using a PRISMA flow diagram (Fig 2). A spreadsheet containing all the included records is provided in S3 Supplement.

### 3.1. Study characteristics

Of the included 196 records, there were 182 (93%) articles published in 115 journals, 10 conference abstracts (5%), and four official health reports (2%). Around 72% (141/196) of included records were published in the second two quarters of the specified timeframe (2012–2023). Most publications were in English (166/196, 85%), followed by French (24/196, 12%), Arabic (4/196, 2%) and Persian (2/196, 1%).

**PRISMA flow diagram including searches of databases and other methods**

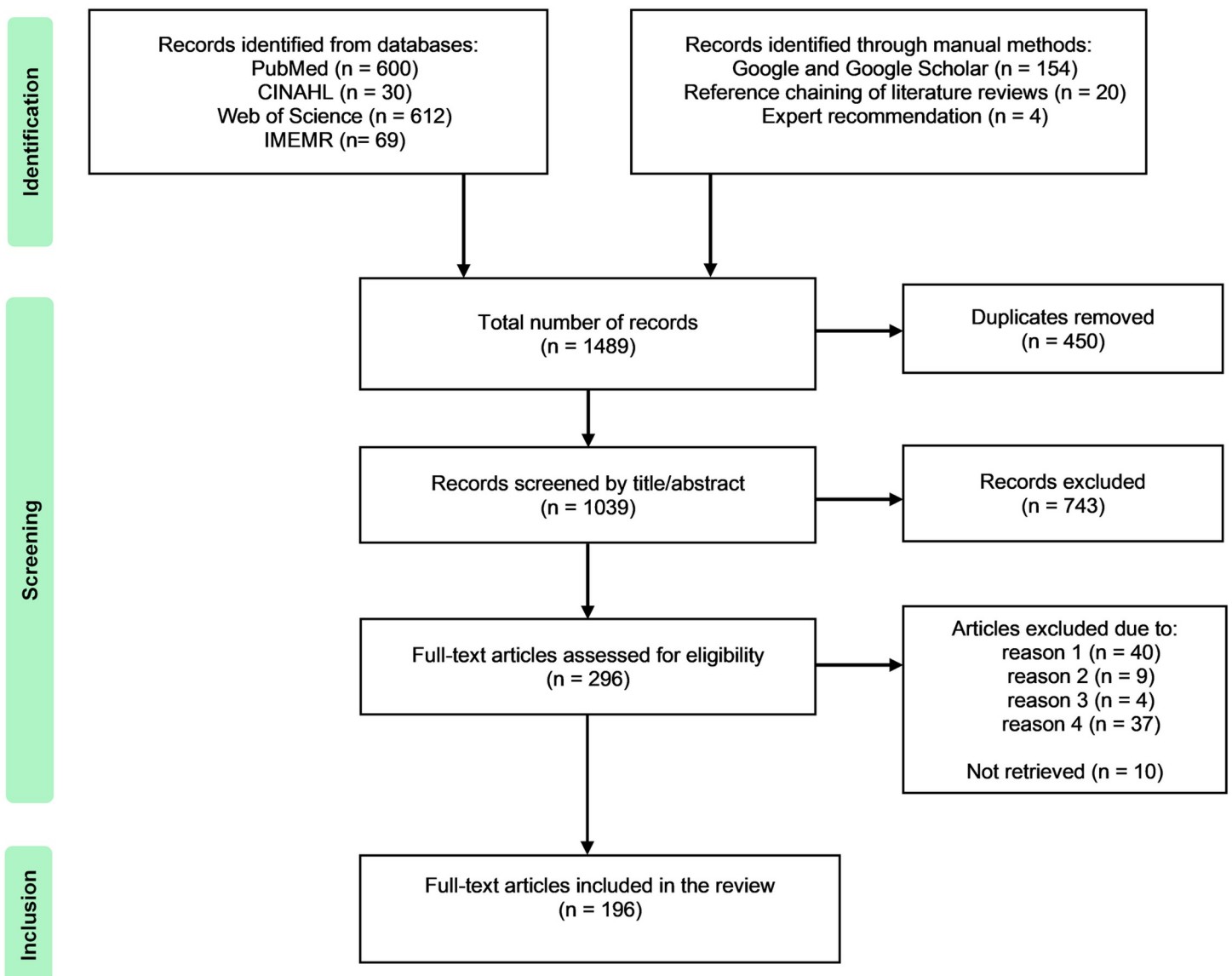

**Fig 2. PRISMA FLOW DIAGRAM OF INCLUDED RECORDS.** Note: Template adapted from Page et al. [33].

More than half (54%) of the publications focused on Iran (37/196, 19%), Morocco (35/196, 18%), and Pakistan (35/196, 18%). This was followed by Saudi Arabia (15/196, 8%), Sudan (13/196, 7%), and Oman (11/196, 6%). While each of the remaining 16 countries had ≤6 articles, we could find only two articles (1%) each for Kuwait, Somalia, and Libya, and no records for Syria or Bahrain. Two studies covered more than one EMR country. Fig 3 shows the number of records per country.

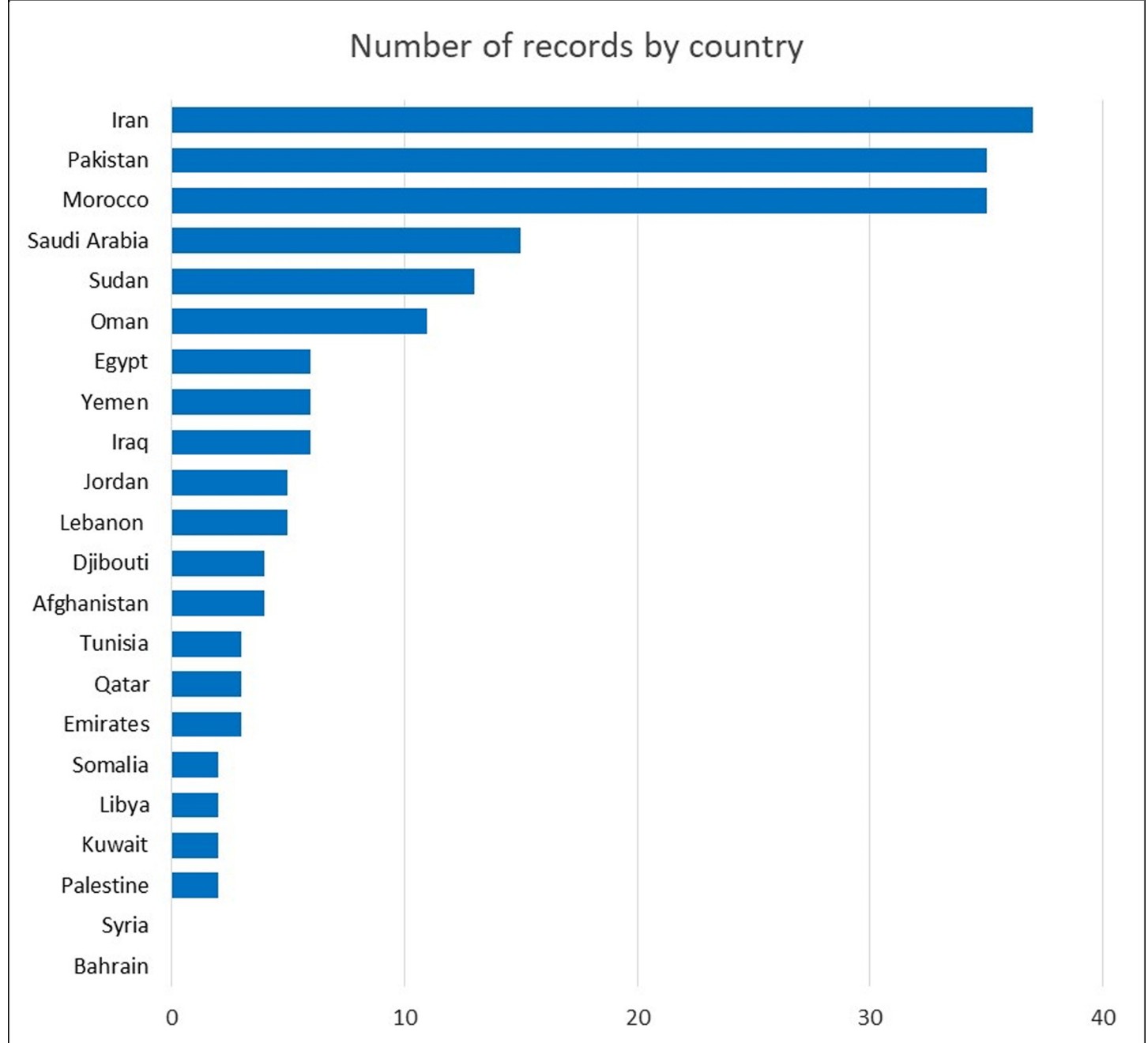

**Fig 3. INCLUDED RECORDS BY COUNTRY FOCUS.**

In terms of methodological focus, a third of included records (67/196, 34%) were case reports or case series. Retrospective and prospective facility-based studies constituted 30% (58/196) and 21% (41/196) respectively. Only 9% (18/196) of publications were national registry-based, and about 4% (7/196) were clinical trials and evaluation studies. Study characteristics are summarized in Table 1.

Authors' affiliations spanned 261 institutes. The Anti-Poison and Pharmacovigilance Center (Morocco), Ibn Tofail University (Morocco) and the Mashhad University of Medical Sciences (Iran) top-ranked with 15, 12, and 9 publications respectively. Around 19% (37/192) of journal and conference papers had at least one author affiliating with an organization outside the region.

### 3.2. Study aims of included records

There were 67 case reports and case series with varied focus and no discernable pattern. For the remaining records, close to half aimed to investigate the clinical and epidemiological features of snake bites irrespective of envenoming status (57/129, 44%), whereas 17% exclusively addressed snakebite envenoming (22/129). Some records had broader or specialized focus with aims spanning across poisoning (12/129, 9%), renal complications (11/129, 9%), animal bites (6/129, 5%), coagulopathy (3/129, 2%), and economic burden of bites (2/129, 2%). There were seven studies (5%) that aimed to examine antivenoms' effectiveness, efficacy, safety, and/or cost.

### 3.3. Burden of snake bites

A total of 170,626 snake bites from 20 countries were reported in papers published between 2000 and 2023. Mortality was not measured in 25 publications (13%), but there were 2,551 deaths reported across the remaining records.

Permanent sequelae were present in 15% (29/196) of included publications. There were another 82 publications (42%) which did not mention permanent sequelae, and thus neither reported nor ruled out these outcomes. The main sequalae that were reported were gangrene and amputations (14 studies); deformed extremities and motor weakness (7 studies); chronic

**Table 1. STUDY CHARACTERISTICS OF INCLUDED RECORDS.**

| Category | | Frequency | Percentage (%) |
|---|---|---|---|
| **Type of study** | Case report/ case series | 67 | 34 |
| | Retrospective facility-based | 58 | 30 |
| | Prospective facility-based | 41 | 21 |
| | National registry-based | 18 | 9 |
| | Trial/evaluation study | 7 | 4 |
| | Other | 5 | 2 |
| **Year of publication** | 2000–2003 | 10 | 5 |
| | 2004–2007 | 14 | 7 |
| | 2008–2011 | 31 | 16 |
| | 2012–2015 | 46 | 23 |
| | 2016–2019 | 52 | 27 |
| | 2020–2023 | 43 | 22 |
| **Language** | English | 166 | 85 |
| | French | 24 | 12 |
| | Arabic | 4 | 2 |
| | Persian | 2 | 1 |

renal damage (3 studies); and visual impairment (3 studies). Other less frequent sequalae were erectile dysfunction and permanent bladder areflexia [34], and chronic post-traumatic stress disorder [35] reported in one record each. Furthermore, three studies collected a disability variable [36–38] with only one study by Akbar et al. providing a working definition of disability as: "development of permanent organ dysfunction (hemiplegia due to intracerebral haemorrhage or permanent loss of higher mental functions due to hypoxic brain damage resulting from respiratory insufficiency)" [36].

Regarding the sociodemographic profiles of snakebite victims, there was a higher frequency in males noted in 97/100 of the population-level records (excluding case reports and case series, and those that did not report on sex). Publications used different ways to report age distribution of victims i.e., mean, median, and frequencies, as well as different age grouping which makes it difficult to summarize. For example, three studies grouped victims as children and adults older than 15 years [39–41]. Snake bites mostly affected rural populace in 39 of the 40 records indicating rural/urban residency. Six studies from Saudi Arabia, Kuwait, and Oman collected data on the nationality of patients i.e., nationals versus non-nationals, with mixed findings from Saudi Arabia [41–43] and Oman [44,45] and higher frequency among non-nationals in one Kuwaiti study [46]. Three prospective studies contained data on education status, showing higher occurrence of snake bites among the least educated subpopulations [47–49]. One study measured socioeconomic status of victims and another reported on family income with findings of higher frequency among those living in poverty or with lower income [38,47].

Six countries in the region collected national-level data on snakebite burden. These were Iran, Saudi Arabia, Palestine, Lebanon, Sudan, and Morocco. The latter four countries covered snake bites in their periodic statistical reports [50–53]. Among the 18 national-level records included in the review, seven calculated an incidence risk. It was lowest in Morocco at 0.34/100,000 for the years 1980–2008 [54]. It was highest in Sudan at 47/100,000 in 2014 [55]. The highest country-wise case fatality rate was observed in Morocco at 5.66% for the years 1992–2007 [56]; however, it decreased to 1.62% in 2021 [53].

## 3.4. Species of offending snakes

The offending snake was not identified in 109 of the 196 included records (56%). The remaining 87 publications proposed identifications for some or all the culprit snakes at family and genus levels (30/87, 34%) or at the species level (57/87, 66%). About 37% of these studies (32/87) did not report the method by which the snake was identified. Those that did specify (55/87, 63%) used different methods to do so, with 13 studies using more than a single method. The most common identification method was body examination of killed or captured snakes in 19 records. Second was victims or witnesses describing or self-identifying the snake in 18 records. Alternatively, snake identification was based on clinical presentation and laboratory findings in 12 articles. Photo catalogues and posters were presented to victims to identify the implicated snakes in seven studies, whereas photography of killed or captured snake were brought by patients in six studies. The bites involved a snake collector, a herpetologist, or a snake charmer in six publications.

Identified indigenous snakes belonged to 30 species. *Cerastes cerastes* were the culprit species in 16 studies and second was *Echis carinatus* in 10 studies. This was followed by *Bitis arietans*, *Echis coloratus*, and *Macrovipera lebetina* in seven records each. *Daboia mauritanica* was cited in six records all from Morocco. Also, *Naja haje* and *Daboia palaestinae* were reported in five and four publications respectively.

Fifteen species were only reported in one record each and almost all were responsible for 1–5 bites. The only exception was *Atractaspis microlepidota* being implicated in 23 bites

according to one study on 63 snakebite-envenomed children from Sudan [57]. There were one sea snake bite (*Hydrophis platurus*) inflicted upon a fisherman in Pakistan [58] and one water snake bite (*Natrix maura*) from Morocco [59]. S4 Supplement provides a table of offending snake species per country as per included records.

Species reported in the literature were consistent with WHO's Snakebite Information and Data Platform and herpetology and toxicology guides except for three snakes. These were *Echis carinatus* from Sudan, *Vipera berus* in Saudi Arabia, and *Cerastes vipera* from United Arab Emirates mentioned in one record each [57,60,61].

## 3.5. Antivenoms and antivenom production centers

Some or all the victims received antivenoms in 75% of included records (147/196), none was administered in 6% (11/196), and it was not clear in the remaining 19% (38/196). Only 64% of records specified the exact number of cases receiving antivenoms (125/196). Half the publications where antivenoms were administered did not provide information on the type (75/147, 51%). For instance, 27 publications only mentioned that the antivenom was 'polyvalent' without providing details on the manufacturer, trade name, and specificity. This lack of information applies to all the records obtained from Sudan, Palestine, Somalia, and Libya.

We counted more than 12 types of antivenoms used in EMR countries. Six types were produced by antivenom production centers from six countries in the Region. These were Pakistan, Iran, Saudi Arabia, Syria, Egypt, and Tunisia. The other types originated from France (FAV-Afrique, Favirept, and *Bitis-Echis-Naja*), Spain (Inoserp MENA), and India (Serum Institute of India polyvalent and VINS Biosnake).

In Iran, the only types of antivenoms documented in the country's literature was produced by the Razi Serum and Vaccine Research Institute [62–67]. This Razi polyvalent antivenin was further used in US-military hospitals in Afghanistan [68,69]. For Saudi Arabia, antivenoms were manufactured by Antivenom and Vaccine Production Center of National Guard health affairs [70–72]. Use of the Saudi antivenoms has also been documented in Oman [44,73–75], Qatar [76,77], United Arab Emirates [75], Kuwait [46], Yemen [78], and US-military hospitals in Iraq [79]. Further, the Syrian 'Antivenom-2' manufactured by Scientific Studies and Research Center was used in Lebanon [80,81], with one study questioning its 'efficacy' in Lebanon as it showed large variation in hospital stays and 67% complications rate requiring intensive care admissions [81].

As for the Tunisian and Egyptian antivenoms, we did not retrieve any records of use outside their countries of origin. In Pakistan, production of the polyvalent produced by the National Institute of Health in Islamabad was insufficient to meet the country's demand, and Indian antivenoms were imported to close the gap [47,82]. The Pakistani antivenom was claimed to be 2.5 times more effective and less costly than its Indian counterpart as per one trial study [82]. Table 2 summarizes data on antivenoms, including types, manufacturers, specificity, countries of use, and number of citing records.

About 27 publications described challenges relating to antivenoms. Shortages in supply were noted in 12 publications, with two studies from Pakistan and Djibouti highlighting that antivenoms were consequently administered below required doses [83,84]. In contrast, antivenoms were over-administered to patients even in absence of "clinical manifestations of envenomation" according to three studies on Yemen [78], and the United Arab Emirates and Oman [75,85]. Furthermore, the antivenoms used were sometimes not specific to the offending species as per nine studies from seven countries. For example, Abu Baker et al. documented the 'inappropriate' use of VINS Biosnake, the only antivenom available in Jordan,

**Table 2. SNAKE ANTIVENOMS IN EASTERN MEDITERRANEAN REGION PER INCLUDED RECORDS.**

| Type of antivenom | Manufacturer | Specificity | Countries of use | Number of records |
|---|---|---|---|---|
| **FAV-Afrique** | Sanofi Pasteur | *Bitis* spp., *Echis* spp., and *Naja* spp. [and *Dendroaspis* spp.] | Morocco, Tunisia, Djibouti, and Iraq | 21 |
| **Razi Antivenin** | Razi Vaccine and Serum Research Institute | *Naja oxiana, Gloydius caucasicus, Echis carinatus, Macrovipera lebetina, Montivipera raddei,* and *Pseudocerastes persicus* | Iran and Afghanistan | 16 |
| **Saudi Guard polyvalent anti-snake venom** | Saudi National Antivenom and Vaccine Production Center (NAVPC) | *Cerastes cerastes, Bitis arietans, Echis carinatus, Echis coloratus, Naja haje* and *Walterinnesia aegyptia* | Saudi Arabia, Oman, Yemen, Qatar, Kuwait, United Arab Emirates and Afghanistan. | 14 |
| **Inoserp MENA** | Inosan Biopharma | *Cerastes cerastes, Echis leucogaster, Bitis arietans, Macrovipera lebetina, Montivipera raddei, Pseudocerastes fieldi, Pseudocerastes persicus, Naja haje, Naja pallida, Naja nigricollis* and *Walterinnesia aegyptia* | Morocco | 7 |
| **Pakistani anti-snake venom** | National Institute of Health Islamabad | *Daboia russelii, Naja naja, Bungarus caeruleus, Echis carinatus* | Pakistan | 4 |
| **Bitis-Echis-Naja** | Pasteur Mérieux | *Bitis, Echis* and *Naja* | Morocco and Djibouti | 3 |
| **VACSERA polyvalent antivenin** | Egyptian Organization Biological products and vaccines (VACSERA) | *Naja haje, Naja nubiae* (incorrectly called *Naja nigricollis*) and *Cerastes cerastes* | Egypt | 3 |
| **Antivenom-2** | Scientific Studies and Research Center | *Daboia palaestinae, Macrovipera lebetina, Montivipera xanthina, Vipera ammodytes,* and *Cerastes cerastes* | Lebanon | 2 |
| **Favirept** | Sanofi Pasteur | *Cerastes cerastes, Echis leucogaster, Bitis arietans, Daboia deserti, Naja nigricollis* and *Naja haje* | Morocco and Afghanistan | 2 |
| **Gamma-VIP** | Pasteur Institute of Tunis | *Cerastes cerastes* and *Macrovipera lebetina* | Tunisia | 1 |
| **SII Snake Anti-venom** | Serum Institute of India Poona | *Daboia russelii, Naja naja, Bungarus caeruleus, Echis carinatus* | Pakistan | 1 |
| **Biosnake** | VINS Andhra Pradesh | *Naja haje, Naja nigricollis* and *Cerastes cerastes*, and manufacturer claims paraspecific activity against 21 other spp. | Jordan | 1 |

against *Atractaspis engaddensis* and *Echis coloratus* concluding that: "Lives could have been saved if specific antivenom had been available" [86].

## 4. Discussion

This scoping review summarized and mapped available evidence on snakebite burden, species and antivenoms using data sought from all 22 EMR countries, and have highlighted existing gaps and challenges. Results indicate a substantial disparity between countries in terms of the volume of publications, and by extrapolation, available data that can identify region-wide information gaps related to snakebite envenoming and data on permanent sequalae, disability, offending species, and types and sources of antivenoms.

From the 196 records included in the review, more than half originated from Iran, Morocco, and Pakistan whilst considerably less were obtained for 16 other countries and it was particularly scarce for those estimated to suffer the greatest burden. For instance, Somalia is estimated to have the highest age-standardized mortality rate from snakebite envenoming in whole of sub-Saharan Africa [2], yet we obtained only two publications which provided little relevant data for that country. Furthermore, we could not retrieve any records from Syria although it is known to have at least seven venomous species [87] and in spite of consulting researchers and experts to locate evidence. As global estimates of snakebite envenoming often rely on extrapolating data from countries with reliable statistics to their neighbors [7], this lack of data in many

EMR countries may negatively skew burden estimates for the EMR and lead to under-estimation of resource needs and prioritization at both domestic and international levels [20].

Disability resulting from snakebite envenoming was poorly characterized and quantified with only three of the included studies attempting to measure disability burdens. As snake bites often take place in rural and low-resource settings without routine follow-up of victims after discharge, the allopathic health systems may fail to capture permanent and long-term disabilities that only become evident post-discharge [88]. For this reason, we need to: (i) look beyond the patients' discharge from the health facility and follow up on long-term and permanent disabilities; and (ii) improve overall reporting on disability and its causes (including snakebite envenoming) so that it does not "slip through the cracks" of existing health information systems.

Annual health reports from the Ministries of Health of Iraq and Oman were excluded because they combined data on snakes with bites and stings from other animals [89,90]. Of note was that the Iraqi report indicated a totality of seven deaths from snake bites and scorpion stings combined in 2022 [90] which is likely an underestimate given the country hosts seven species of venomous snakes [87] and 25 species of scorpions [91] interspersed with a large human population at risk. This review identified three single-center records from Iraq reporting alarmingly high case fatality rates of 7.1%, 19.7% and 28.6% [92–94] for snake bites. While functional routine health information systems would generate more realistic figures for such countries, they will unlikely suffice to reflect the full burden with considerable proportions of affected populations seeking treatment from traditional healers [95,96]. It is thus vital to complement health system-acquired data with community-based surveillance [4].

Most publications (~70%) did not identify the culprit species. We realize that snake identification is challenging worldwide [29]. However, what we find concerning are: (i) instances where dead snakes were brought to the health facility yet overlooked in hospital records [44,75]; and (ii) arguments that species identification becomes unimportant in settings where only one type of polyvalent antivenom is available [75,97]. When feasible, information about the species of snake responsible can be useful to informing clinical management plans in the event that clear and demonstrable signs of envenoming arise. Information about biting species is also useful for documenting clinical syndromes of envenoming and mapping the incidence of species-specific cases of envenoming. However, identification of a species of venomous snake as a culprit in the absence of clear signs of clinical envenoming should not be used to justify administration of antivenom, since many snakes produce "dry bites" where no venom is actually injected [98]. The clinical decision to administer antivenoms must always be based upon a careful clinical history and the elucidation of evidence of clinical signs of envenoming [15]. Data about which species are responsible for snakebites can be useful in evidence-informed decision-making on the type of antivenoms to be procured for future use in a given location.

We counted nine records from seven countries where the procured antivenoms were not specific to indigenous snakes e.g., one paper from Jordan reported use of an Indian antivenom with no proven paraspecificity against any of the country's species [86]. Procurement agencies should establish clear criteria for the selection of antivenoms which is based on proven efficacy against the snake species that occur in the countries or region concerned.

Included records in our review were mostly consistent with both herpetology and toxicology guides on the Region [30–32], and WHO's Snakebite Information and Data Platform [99]. Expectedly, there were species listed in such guides and the database but were not reported in the literature. This can be explained by the fact that there were no or few studies from several countries in the Region and that most publications did not identify the offending species.

Further, the extent to which humans are exposed to snake bites depends on the extent to which human activity overlaps with snake habitat.

Based on the literature, it appears that regional collaboration on snakebite envenoming issues is very limited. As neighboring countries often face common venomous species and antivenom supply challenges, enhancing regional collaboration and partnerships is likely to benefit all stakeholders [4]. Shared learning and policy transfer are possible with examples such as the Moroccan control strategy's success in increasing snakebite notification by 150% [100] and their Anti-Poison and Pharmacovigilance Center's role in collecting and collating routine data from across the country [101]. The fact that six EMR countries aggregate national-level data is promising and clearly demonstrates that integrating data on snake bites into routine health information systems is doable even in conflict-affected and developing contexts. WHO provides templates for reporting of snakebite data through the DHIS2 (https://: dhis2.org) platform. Countries would also benefit from regional coordination to incorporate snakebite prevention and control into existing regional strategies and health programs like malaria and hookworms [102].

There have been some traditional literature reviews in the Region [20,87,103] but the systematic and comprehensive approach of the scoping review methodology constitute a defining strength of this review. In addition, the search strategy covered all 22 EMR countries, consulted numerous electronic databases and manual methods, and eventually excluded no records based on language. We thus believe that we exhausted the greatest share of the available literature. The review does, however, have its limitations. First, study selection and data extraction were largely performed by one reviewer. We ameliorated this by seeking a second reviewer in case of uncertainty and performing regular checks of extracted data; however, there was some possibility of error. Second, studies covered overlapping timespans and some had referred to the same data sources which could have led to double counting in some countries. This would have only affected countries with national registries and not the majority of countries with very limited publications.

## 5. Conclusion

This scoping review summarized and mapped evidence on snakebite burden, species and antivenoms in the Eastern Mediterranean. Despite the relatively large number of publications identified, evidence on snake bites and snakebite envenoming remains limited to absent for most Eastern Mediterranean countries, and particularly for those estimated to bear a great burden. Disability was poorly characterized and quantified in all countries constituting a notable gap for future research to address. Action is needed at national and regional levels to enhance collection, collation and analysis of routine and surveillance data; regional collaboration; and further research beyond single health-facility or case report studies. These would be pivotal in order to build the evidence base to address snakebite envenoming in the Eastern Mediterranean.

## Supporting information

**S1 Supplement. Preferred Reporting Items for Systematic Reviews and Meta-Analyses extension for Scoping Reviews (PRISMA-ScR) Checklist.**
(DOCX)

**S2 Supplement. Search strategies.**
(DOCX)

**S3 Supplement. Spreadsheet of all included records and extracted data.**
(XLSX)

**S4 Supplement. Offending species by country.**
(PDF)

# Acknowledgments

We gratefully acknowledge Dr. Mohammed Saadati for his help in extracting data from studies in Persian; Dr. Arash Rashidian for his feedback on the study; Ms. Gehane Al Garraya for her support with the search strategy and retrieving full text articles; Ms. Khushbu Gupta, Dr. Anna Pintor, Mr. Cameron Denney and Ms. Kathleen Krupinski for sharing their knowledge and advising on the map; and Dr. Henry Doctor, Dr. Supriya Warusavithana, and Dr. Mona Osman for their support throughout the project.

**Disclaimer**

The authors alone are responsible for the views expressed in this article and they do not necessarily represent the views, decisions or policies of the institutions with which they are affiliated.

# Author Contributions

**Conceptualization:** Ali Alshalah, David J. Williams, Alessandra Ferrario.

**Formal analysis:** Ali Alshalah.

**Investigation:** Ali Alshalah.

**Methodology:** Ali Alshalah, David J. Williams, Alessandra Ferrario.

**Project administration:** Alessandra Ferrario.

**Resources:** Alessandra Ferrario.

**Supervision:** David J. Williams, Alessandra Ferrario.

**Validation:** Ali Alshalah, Alessandra Ferrario.

**Visualization:** Ali Alshalah, Alessandra Ferrario.

**Writing – original draft:** Ali Alshalah.

**Writing – review & editing:** David J. Williams, Alessandra Ferrario.

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
