## [Decision Letter · Decision Letter 0]

8 Dec 2023

Dear Dr Alshalah,

Thank you very much for submitting your manuscript "From fangs to antidotes: a scoping review on snakebite burden, species, and antivenoms in the Eastern Mediterranean Region" for consideration at PLOS Neglected Tropical Diseases. As with all papers reviewed by the journal, your manuscript was reviewed by members of the editorial board and by several independent reviewers. In light of the reviews (below this email), we would like to invite the resubmission of a significantly-revised version that takes into account the reviewers' comments. 

Both reviewers identified the need of this manuscript due to the neglect of research on envenoming in this region. Both reviewers have suggested substantial, mostly editorial, comments to improve the flow and interpretation of the manuscript for a potential reader. I recommend you apply these suggestions in re-draft for resubmission. 

We cannot make any decision about publication until we have seen the revised manuscript and your response to the reviewers' comments. Your revised manuscript is also likely to be sent to reviewers for further evaluation.

Sincerely,

Stuart Robert Ainsworth

Academic Editor

José María Gutiérrez

Section Editor

Reviewer's Responses to Questions

**Key Review Criteria Required for Acceptance?**

**Methods**

-Are the objectives of the study clearly articulated with a clear testable hypothesis stated?

-Is the study design appropriate to address the stated objectives?

-Is the population clearly described and appropriate for the hypothesis being tested?

-Is the sample size sufficient to ensure adequate power to address the hypothesis being tested?

-Were correct statistical analysis used to support conclusions?

-Are there concerns about ethical or regulatory requirements being met?

Reviewer #1: See report

Reviewer #2: The objectives of this study are clearly defined and the method design appropriate. Limitations to the methodology are also acknowledged in the discussion.

**Results**

-Does the analysis presented match the analysis plan?

-Are the results clearly and completely presented?

-Are the figures (Tables, Images) of sufficient quality for clarity?

Reviewer #1: See report

Reviewer #2: Results are clearly presented. It would be nice to have another figure or two displaying results, but this is understandable as this is not an experimental study but a literature review.

**Conclusions**

-Are the conclusions supported by the data presented?

-Are the limitations of analysis clearly described?

-Do the authors discuss how these data can be helpful to advance our understanding of the topic under study?

-Is public health relevance addressed?

Reviewer #1: See report

Reviewer #2: Conclusions are supported by the data and its relevance to public health is addressed.

**Editorial and Data Presentation Modifications?**

Reviewer #1: See report

Reviewer #2: This manuscript is well written and provides a good summary of snakebite in the Mediterranean region, which has received considerably less attention than other areas of the world in regard to snakebite. I recommend this manuscript for publication after minor revision.

**Summary and General Comments**

Reviewer #1: The following points should be addressed:

• Define Eastern Mediterranean countries; this can be in the form of a map.

• In the Supplement IV: Offending Species by country, the authors did not include Atractaspis, there are cases attributed to this snake in Jordan and Saudi Arabia. Also cases caused by Pseudocerastes fieldi were recorded from Jordan 

• Citations of some references should be checked such as:

Amr ZS, Abu Baker MA, Warrell DA. Terrestrial venomous snakes and snakebites in the 582 Arab countries of the Middle East. Toxicon Off J Int Soc Toxinology. 2020 Apr 15;177:1–15.

• Some important references on Jordan and Qatar are missing:

Saadeh, A.M., 2001. Case report: acute myocardial infarction complicating a viper bite. Am. J. Trop. Med. Hyg. 64, 280–282. https://doi.org/10.4269/ajtmh.2001.64.280 -for Jordan

Elmoheen, A., Salem, W. A., Haddad, M., Bashir, K., Thomas, S. H. 2020. Experience of Snakebite Envenomation by a Desert Viper in Qatar. Journal of Toxicology. https://doi.org/10.1155/2020/8810741

• In the Supplement IV: it lists 15 species of snakes, while the authors stated that 30 species were Identified indigenous snakes.

• The time specified between 200-2023 caused some discrepancy in the other species that are considered venomous such as Naja sp. in both the Middle East and North Africa. At least these species should be mentioned. 

Conclusion

First, I thank the authors for their efforts to outline this review, it is very important for all health officials of countries in the study area. However, the presentation of the results should be improved to facilitate the reader to better understand a sequential, ordered and graphical results.

Graphs, maps and tables that can be extracted from the supplementary data should be incorporated to enhance the final format. 

The manuscript should be revised as suggested.

Reviewer #2: The manuscript "From fangs to antidotes: a scoping review on snakebite burden, species, and antivenoms in the Eastern Mediterranean Region is well written and reports on a region that has been neglected in regard to reports on snakebite. My main recommendation are as follows: 

1. At least one other figure should be added to help readers visualise the results, as there are few figures in this manuscript. A figure depicting the region studied on a map (all countries evaluated) would add to the manuscript, or could even be an additional panel in Figure 2. 

2. Additionally, maybe incorporation of information from venomous snake field guides in this region, it would be good to know what is being reported in the scientific literature vs information from herpetology guidebooks. 

3. Lastly, this manuscript might be better considered a systematic review and not a primarily research article?

PLOS authors have the option to publish the peer review history of their article (what does this mean?). If published, this will include your full peer review and any attached files.

Reviewer #1: No

Reviewer #2: No

Figure Files:

Data Requirements:

Please note that, as a condition of publication, PLOS' data policy requires that you make available all data used to draw the conclusions outlined in your manuscript. Data must be deposited in an appropriate repository, included within the body of the manuscript, or uploaded as supporting information. This includes all numerical values that were used to generate graphs, histograms etc.. For an example see here: http://www.plosbiology.org/article/info:doi%2F10.1371%2Fjournal.pbio.1001908#s5.
---

## [Editor Report · Decision Letter 1]

7 May 2024

Dear Dr Alshalah,

We are pleased to inform you that your manuscript 'From fangs to antidotes: a scoping review on snakebite burden, species, and antivenoms in the Eastern Mediterranean Region' has been provisionally accepted for publication in PLOS Neglected Tropical Diseases.

Best regards,

Stuart Robert Ainsworth

Academic Editor

José María Gutiérrez

Section Editor

---

## [Editor Report · Acceptance letter]

4 Jun 2024

Dear Dr. Alshalah,

We are delighted to inform you that your manuscript, "From fangs to antidotes: a scoping review on snakebite burden, species, and antivenoms in the Eastern Mediterranean Region," has been formally accepted for publication in PLOS Neglected Tropical Diseases.

Best regards,

Shaden Kamhawi

co-Editor-in-Chief

Paul Brindley

co-Editor-in-Chief
